# Decarbonization potential of electrifying 50% of U.S. light-duty vehicle sales by 2030

Maxwell Woody [1] ✉, Gregory A. Keoleian [1] & Parth Vaishnav[1]

The U.S. federal government has established goals of electrifying 50% of new light-duty vehicle sales by 2030 and reducing economy-wide greenhouse gas emissions 50-52% by 2030, from 2005 levels. Here we evaluate the vehicle electrification goal in the context of the economy-wide emissions goal. We use a vehicle fleet model and a life cycle emissions model to project vehicle sales, stock, and emissions. To account for state-level variability in electric vehicle adoption and electric grid emissions factors, we apply the models to each state. By 2030, greenhouse gas emissions are reduced by approximately 25% (from 2005) for the light-duty vehicle fleet, primarily due to fleet turnover of conventional vehicles. By 2035, emissions reductions approach 45% if both vehicle electrification and grid decarbonization goals (100% by 2035) are met. To meet climate goals, the transition to electric vehicles must be accompanied by an accelerated decarbonization of the electric grid and other actions.

The urgent need to reduce greenhouse gas (GHG) emissions is leading to major changes in the transportation sector[1]. The most prominent strategy for decarbonizing transportation is electrifying light-duty vehicles (LDVs)[2], which account for approximately 75% of passenger miles traveled and 50% of transportation sector GHG emissions in the U.S.[3] It is estimated that battery electric vehicles (EVs) will achieve upfront cost parity with conventional vehicles by 2030–2035[4] and many automotive companies have plans for rapid electrification in the next decade (Table S1). In addition to reducing GHG emissions, large-scale EV adoption may have significant local air quality and human health co-benefits[5]. Through an Executive Order, the U.S. has set a nonbinding target for 50% of LDV sales to be electric by 2030[6]. New fuel economy standards proposed by the Environmental Protection Agency (EPA) may result in 67% of new LDV sales being electric by 2032[7]. And some U.S. states, led by California, have a more ambitious target for 100% of LDV sales to be electric by 2035[8]. To fully realize the carbon reduction benefits of EVs, decarbonizing the electric grid and preparing for increased electricity demand are critical[9,10]. As the grid GHG intensity varies across the country, so too does the emissions impact of vehicle electrification[11]. Here we develop a state-by-state model of the U.S. LDV fleet, quantify the impact of LDV electrification on GHG emissions, and evaluate the emissions reduction in relation to short-

term (2030) Intergovernmental Panel on Climate Change (IPCC) carbon reduction targets and U.S. climate goals.

Prior models of U.S. LDV fleet GHG emissions have been conducted on a national level and have focused on 2050 targets[12–14]. Alarfaj et al. note that 80–90% decarbonization is possible by 2050, but this may require reductions in vehicle miles traveled (VMT) in addition to electrification[13]. Milovanoff et al. found that 90% of the on-road fleet would need to be electrified by 2050 to be consistent with 2 °C climate targets[14]. Zhu et al. found that 100% of LDV sales would need to be electric by 2040 at the latest to stay within a 2 °C target[12]. All three note the difficulties posed by long fleet turnover timelines, and the requirement for both vehicle electrification and electric grid decarbonization.

These studies each use a bottom-up approach, calculating fleet emissions under a variety of scenarios in order to determine targets (e.g., 90% electrification by 2050). We use a top-down approach based on stated goals and proposed regulations (50% electrification of sales by 2030; 67% by 2032) along with current trends to determine what the emissions would be if those goals are met. Our base scenario is defined by reaching exactly 50% EV sales nationwide in 2030. This results in 69% EV sales nationwide in 2032, so our base scenario is consistent with both stated goals (50% in 2030)[6] and proposed EPA regulations (67% in 2032)[7]. Modeling each state individually, rather than the

[1]Center for Sustainable Systems, School for Environment and Sustainability, University of Michigan, Ann Arbor, MI 48104, USA.
✉e-mail: maxwoody@umich.edu

country as a whole, is a novel contribution of our study that allows us to investigate the impact of state-level heterogeneity in EV adoption levels and grid emissions factors on overall LDV emissions. As the proportion of EVs in the fleet grows, where and when these vehicles charge will become more important[15]. There are currently 15 states that have adopted a more aggressive sales goal of 67% by 2030 (rather than the national goal of 50%), and 100% by 2035 under the Clean Air Act Section 177[16]. Our base model results in a weighted average EV sales percentage of 64.4% by 2030 and 94.2% by 2035 in these states, showing that our base case reasonably captures multiple Federal policies (Biden Administration goals and proposed EPA regulations) and state policies (ZEV goals under CAA 177). We include more aggressive grid decarbonization scenarios, in line with updated U.S. government targets[6]. We incorporate upstream emissions from vehicle fuels and electricity and vehicle production, which are not uniformly included in transportation sector models. And this study focuses on short-term (2030) goals, though we calculate emissions out to 2035.

In addition to the major trends of vehicle electrification and grid decarbonization, there are complementary and competing trends that contribute to LDV fleet emissions. These include improvements in fuel economy for EVs and internal combustion engine vehicles (ICEVs), the shift away from cars in favor of light trucks[17], and changes in the number of vehicles sold each year.

Here we use different grid development scenarios to explore the limits of what vehicle electrification can accomplish for decarbonization and consider if the current U.S. vehicle electrification goal by itself is sufficient to meet U.S. climate goals. We highlight additional decarbonization strategies including reducing vehicle size, reducing VMT, and accelerating retirement of vehicles. We find that meeting the U.S. vehicle electrification sales goal along with fleet turnover results in roughly 25% reduced emissions for the LDV fleet by 2030, well short of the 50–52% economy-wide emissions reduction goal. However, by 2035, reductions approach 45% when vehicle electrification is combined with rapid grid decarbonization. Therefore, the ongoing transition to electric vehicles must be accompanied by an accelerated decarbonization of the electric grid and augmented by additional actions such as decarbonization of liquid fuels, reducing travel demand, shifting to less carbon-intensive modes of transportation (e.g., mass transportation), and accelerating fleet turnover through early retirement of ICEVs.

## Results

As part of its Nationally Determined Contribution to the Paris Climate Accords, the U.S. set an interim target to reduce emissions economy-wide by 50–52% by 2030, compared to 2005 levels[18]. The 2030 economy-wide target includes a grid decarbonization target of 100% by 2035 but does not include specific goals for the transportation sector. For simplicity, we use a proportional emissions reduction as a point of comparison in our analysis. It is widely accepted that LDVs will be less difficult to decarbonize than medium-duty and heavy-duty vehicles, shipping, and aviation[19]. Thus, it seems reasonable to assume that to meet national GHG emission reduction goals a proportional share of transportation emissions reductions is the very least that should be expected of LDVs[20]. We determine a baseline value of 1600 Mt $CO_2e$ in 2005 by adjusting EPA data[21] to account for upstream emissions of fuels and vehicle production emissions (materials, manufacturing, and vehicle disposal). Though vehicle production emissions are not included in the EPA reporting, they are important to consider for fleet analyses because as the fleet electrifies and the grid decarbonizes, vehicle production emissions will make up a greater proportion of total fleet emissions[12]. Therefore, we compare our results with an aspirational target of 800 Mt $CO_2e$ in 2030 (50% of 2005 emissions), while noting that the U.S. has no formal LDV emissions goal for 2030. This value (800 Mt) is also a proportionate contribution to IPCC global targets (45% reduction in emissions by 2030 from 2010 levels)[22]. Given its historic emissions and economic and technological leadership, it is reasonable to expect that the U.S. would go beyond a proportional contribution.

### Vehicle sales, stock, and emissions

We model the growth rate of the EV share of LDV sales as a symmetrical logistic curve ("S-curve"). Logistic curves are used in a wide range of technology adoption and technology diffusion studies[23], including for EVs[24,25]. We use the same growth rate for each state in the U.S., but with different initial conditions based on 2021 sales (see Methods). We tune the growth rate such that the target value of 50% EV sales nationwide in 2030 is reached exactly. This results in 69% EV sales nationwide in 2032, so our model is consistent with both stated goals (50% in 2030)[6] and proposed EPA regulations (67% in 2032)[7]. Each state has unique adoption curves. For example, California would reach 79% EV sales in 2030, while North Dakota would reach 16% EV sales in 2030 (Fig. 1). The percentage of EVs is also broken down by cars and trucks (e.g, 93% of cars and 71% of trucks in California, 39% of cars and 11% of trucks in

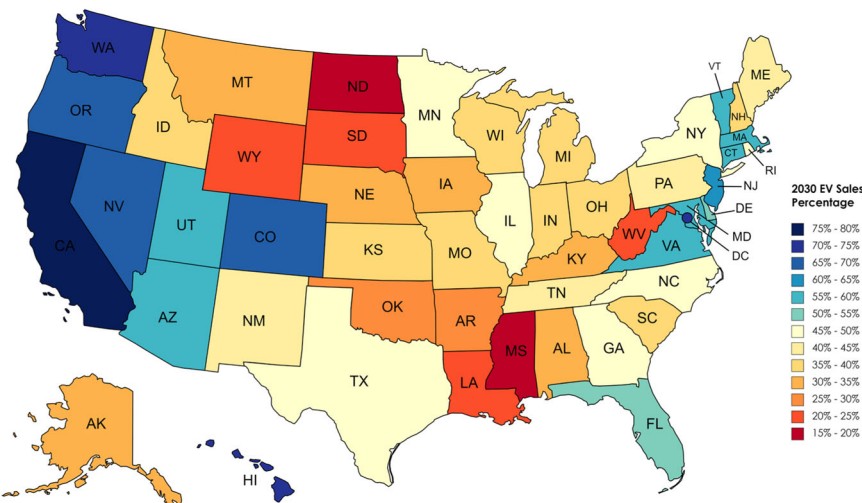

**Fig. 1 | Projected electric vehicle sales percentage in each state in 2030.** Projected EV sales in each state, with a national average sales percentage of 50% in 2030, based on the current distribution of EV sales by state. Created with mapchart.net. https://www.mapchart.net/terms.html#licensing-maps.

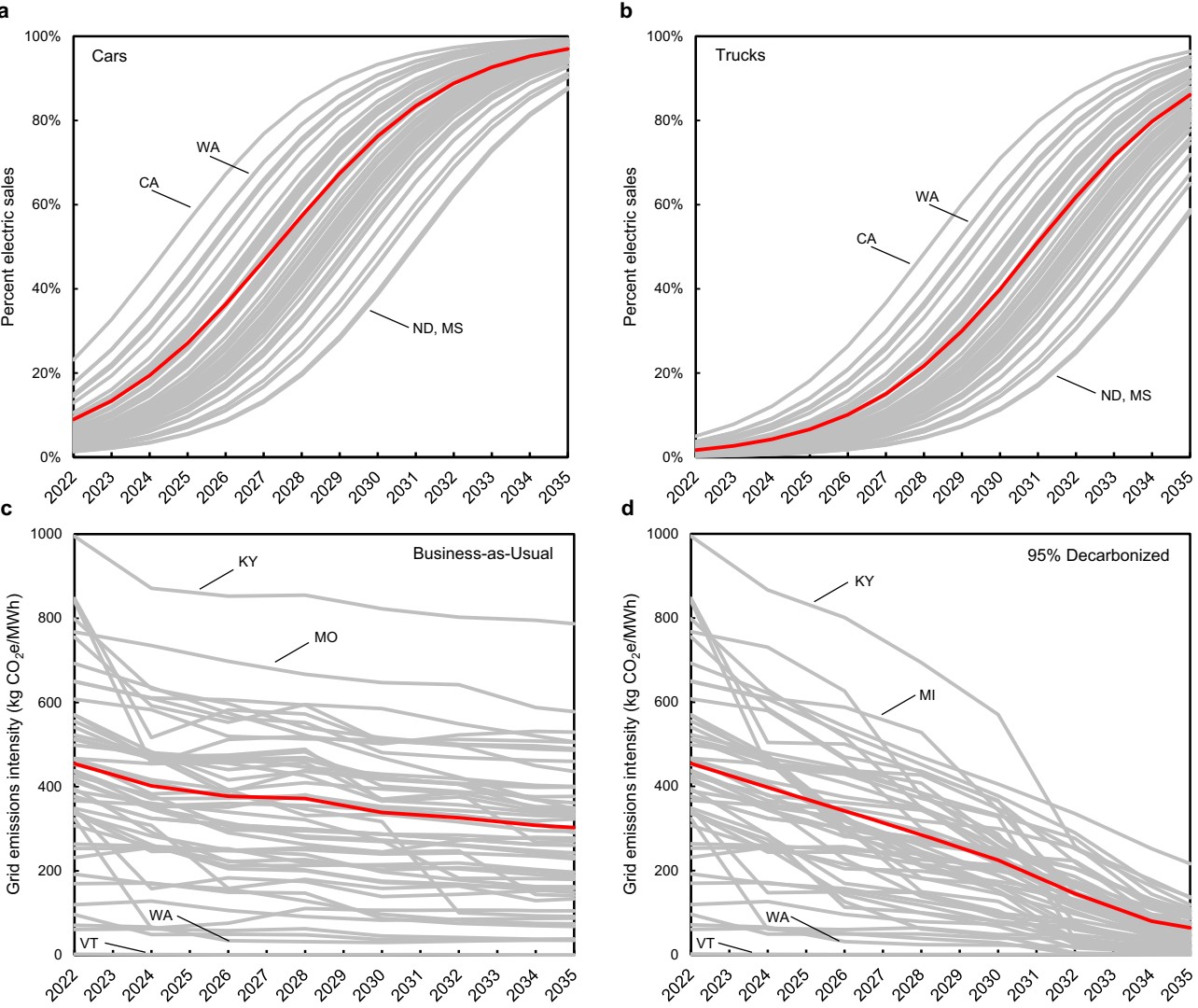

**Fig. 2 | Annual *state-by-state* electric vehicle sales and grid emissions intensity projections.** Annual state-by-state electric vehicle sales percentages for (**a**) cars and (**b**) trucks. Annual state-by-state electric grid emissions intensities for (**c**) the business-as-usual grid scenario and (**d**) the 95% decarbonized electricity by 2035 grid scenario. Highest and lowest states are labeled; red lines are U.S. national average values.

North Dakota) (Fig. 2a, b). Nationally, reaching 50% electric sales for all LDVs in 2030 is accomplished here by electrifying approximately 75% of car and 40% of truck sales.

The 50% electrification rate for new vehicle sales in 2030 has a limited impact on the vehicle stock in that year, due to the time required for the LDV fleet to turn over. The median vehicle in the U.S. is on the road for approximately 20 years[26]. Using vehicle survival curves from the Transportation Energy Data Book[27], we show that achieving a 50% electric sales rate in 2030 would lead to 11.5% of the LDV stock being electric in 2030. This percentage could be even lower if vehicle lifetimes continue to increase[26]. The shape of the adoption curve has a limited impact; even if sales were to grow linearly to 50%, rather than logistically, only 15.1% of the LDV stock would be electric in 2030.

We extend our analysis to 2035 to investigate impacts farther in the future from achieving the 2030 sales target. In 2035 our model has an 89% EV sales rate and 31% of the overall vehicle stock is electric. For both 2030 and 2035 these values are higher for cars and lower for trucks (Fig. 3a). We also note that the average annual VMT decreases throughout a vehicle's lifetime. Therefore, assuming similar driving patterns across powertrains, by 2035 the 31% of vehicles that are electric would be responsible for 37% of all miles driven by LDVs (Fig. 3b). Our vehicle fleet model also has cars declining from 33% to

28% of sales from 2022 to 2035, and a corresponding decrease of cars in the vehicle stock from 40% to 29% (Fig. 4).

To calculate fleet emissions, we use two electric grid scenarios from the NREL Cambium model[28]. The first is a business-as-usual scenario with modest progress towards decarbonization (Fig. 2c). The second includes 95% decarbonization of the electric grid by 2035, from 2005 levels, approximating the U.S. goal of 100% grid decarbonization by 2035 (Fig. 2d)[6]. The emissions associated with a 50% electrification rate for 2030 new LDV sales with the business-as-usual grid would be 1210 Mt in 2030, a 24% reduction from 2005 levels, falling well short of the 50% U.S. reduction target (Fig. 4, Fig. 5a). With 95% decarbonization of the grid by 2035, LDV emissions in 2030 would be 1190 Mt (Fig. 5c). The impact of grid decarbonization is initially limited by the relatively small number of EVs in the on-road fleet. In 2030, regardless of how decarbonized the electric grid has become, approximately 90% of LDV emissions are from ICEVs. In 2035 the impact of rapid grid decarbonization can be seen more clearly, as many more EVs are on the road (Fig. 5b, d).

## Attribution of emissions reductions

In our base case, most of the initial improvement (between the present and 2030) comes from fleet turnover. Here fleet turnover is quantified

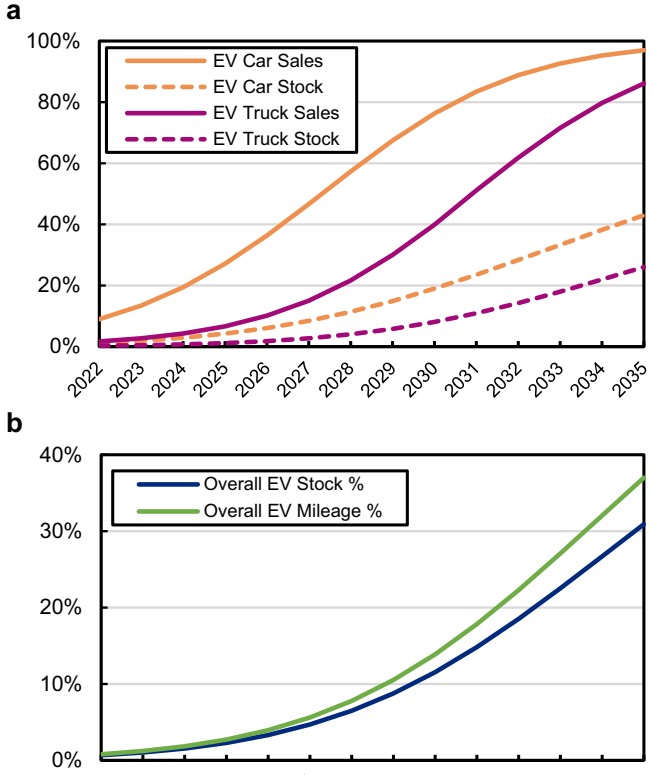

**Fig. 3 | Electric vehicle sales, stock, and mileage percentages.** Model results showing (**a**) national EV sales (solid lines) and stock (dashed lines) percentages for cars (orange) and trucks (red) through 2035, (**b**) national EV stock percentage (blue) and EV mileage percentage (green) through 2035, in the baseline scenario.

as the reduction in emissions from replacing older vehicles (at their natural end-of-life) with new vehicles of the same powertrain (but with improved fuel economy), as well as minor changes in the overall fleet size. This does not include changes in vehicle electrification or grid decarbonization from 2022 levels. In other words, if the electrification rate remained at 4% of sales and grid emissions factors remained the same as in 2022, the emissions of the LDV fleet would be 13% (183 Mt) lower in 2030 than in 2022 (Fig. 5a). The trend away from cars and towards trucks is responsible for a small (10 Mt) increase in emissions. The impact of reaching a 50% EV sales share in 2030, with business-as-usual grid development, is a 5% (62 Mt) reduction in vehicle use phase emissions. Some of this reduction (25 Mt) is offset by increased vehicle production emissions, as EVs have higher production emissions than ICEVs[11].

When the analysis is extended to 2035 (Fig. 5b), vehicle electrification, responsible for a 3% reduction by 2030, is responsible for a 11% (161 Mt) reduction in emissions by 2035. The impact of electrification is amplified by the level of grid decarbonization. Business-as-usual grid decarbonization, without increases in EV sales, would result in only a 5 Mt decrease in LDV emissions by 2035; reaching the 50% EV sales target, without any changes to the grid, would result in a 116 Mt decrease in use-phase LDV emissions. When these factors are combined the decrease is 193 Mt, showing that the impact of vehicle electrification and grid decarbonization together is greater than the sum of each individual factor in isolation.

## Grid decarbonization

Comparing the business-as-usual grid scenario (Fig. 5 a, b) with the 95% decarbonization by 2035 grid scenario (Fig. 5 c, d) reveals that more ambitious grid decarbonization policies can substantially reduce transportation sector emissions, though the overall impact will not be large until after 2030, when there is a greater percentage of EVs in the fleet. The compounding effect of grid decarbonization and vehicle electrification can be observed by comparing Fig. 5b and d. In 5d, the

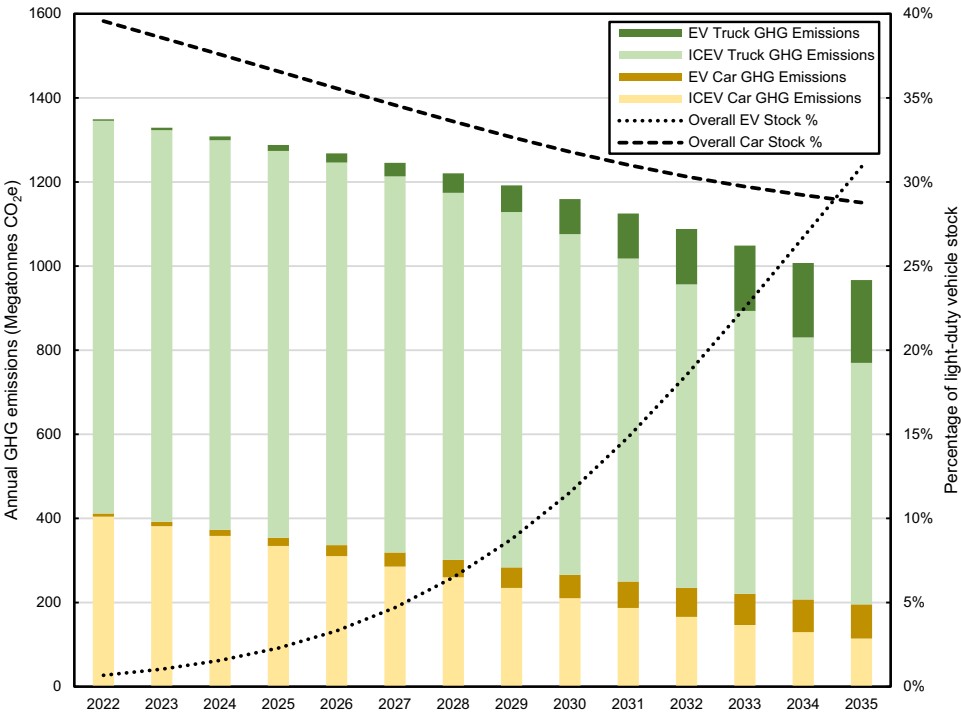

**Fig. 4 | Annual national greenhouse gas emissions in the baseline scenario by source.** Annual national greenhouse gas emissions from ICEV cars (light yellow), ICEV trucks (light green), and EV cars (dark yellow), and EV trucks (dark green); the percentage of vehicles in the fleet that are electric (dotted line), and the percentage of vehicles in the fleet that are cars (dashed line), in the baseline scenario.

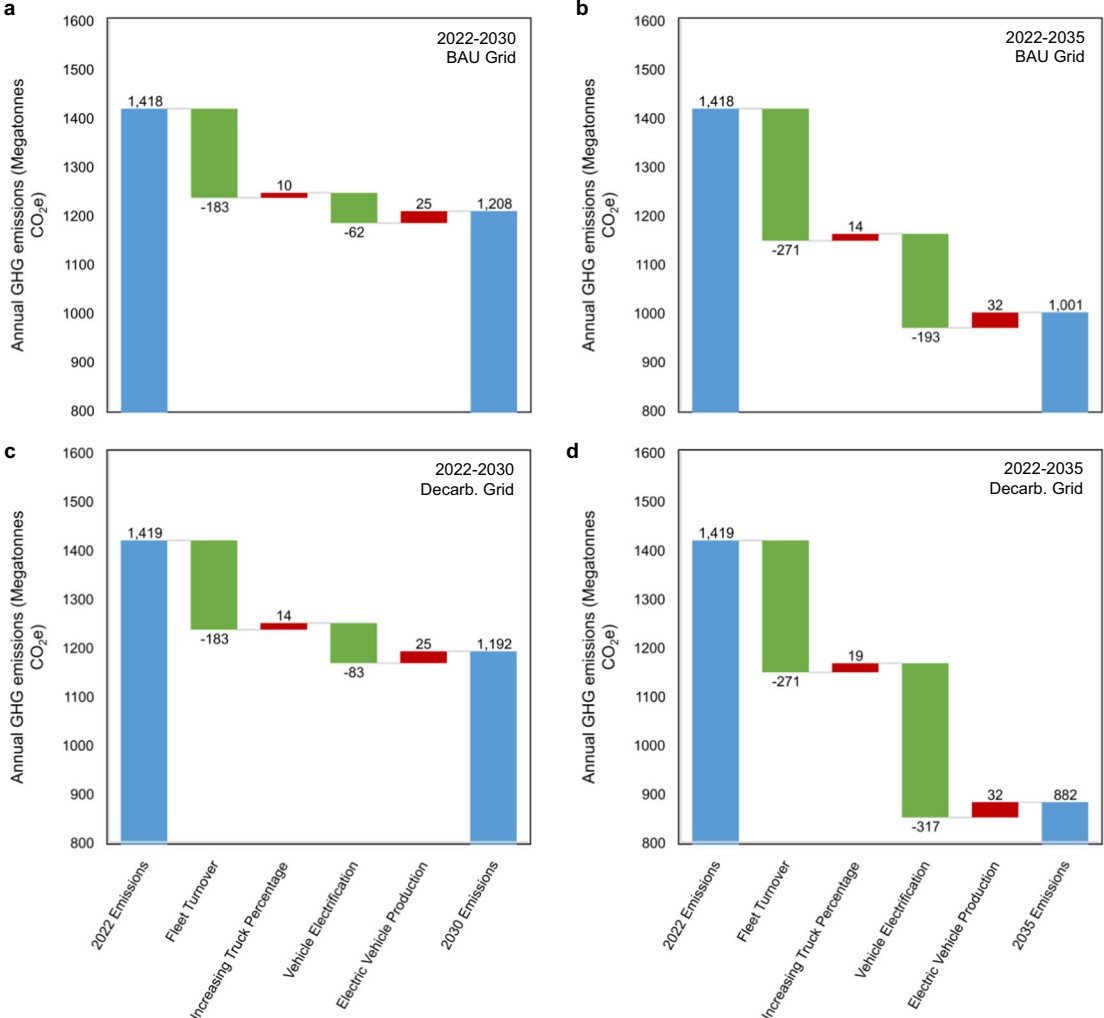

**Fig. 5 | Attribution of greenhouse gas emissions reductions in different grid scenarios.** Attribution of greenhouse gas emissions reductions (**a**) business-as-usual (BAU) grid between 2022 and 2030, (**b**) BAU grid between 2022 and 2035 (**c**) 95% decarbonized grid between 2022 and 2030, (**d**) 95% decarbonized grid between 2022 and 2035. Green bars represent decreases in emissions while red bars represent increases in emissions.

emissions reduction from improving the grid (absent any changes in EV sales) would be 11 Mt (5 Mt in 5b). The emissions reduction from vehicle electrification, without any changes to the grid, would be 116 Mt in 5d (the same as 5b). Yet combining these two factors leads to a decrease of 317 Mt CO₂e (193 in 5b). In this scenario, with 95% grid decarbonization by 2035, and EVs reaching 50% of sales by 2030, the LDV fleet approaches a 45% emissions reduction, from 2005 levels, in 2035.

### State-level considerations

We hypothesized that national fleet models may underestimate the emissions reduction enabled by electrification if states that electrify their vehicle fleets faster than the U.S. average (Fig. 2 a, b) also have below average grid emissions intensities (Fig. 2 c, d). However, with the 50% electrification in 2030 target, the difference in GHG emissions between our state-level model, and our model using the national average grid in every state is less than 2% each year through 2035. If the grid decarbonizes more rapidly, the difference in models is less than 0.5% (Fig. 6). This small difference again reflects the fact that most emissions through 2030 will be from ICEVs. Expressed as a percentage of EV emissions, rather than total emissions, the state-level model results in EV emissions that are up to 7% lower than the national model, using a business-as-usual grid, and

up to 4% lower using the decarbonized grid. This suggests that there may be a limited time in which there is an opportunity to lower emissions by concentrating EV deployment in certain states, but as the grid decarbonizes more fully this regional variability, and the opportunity to reduce emissions by strategic deployment, would diminish.

### Battery material constraints

Under our base scenario, the US EV stock reaches 11.5% in 2030 and 30.8% in 2035. This is approximately 31 million EV cars and 47 million EV trucks on the road in 2035. This would require roughly 7.5 TWh of batteries, using current vehicle battery sizes. This is roughly equal to the total battery manufacturing capacity using U.S. lithium reserves, but well below the total manufacturing capacity with global lithium reserves (209 TWh)[29]. The manufacturing capacity using U.S. reserves of other critical materials are more limited (0.7 TWh for cobalt and 0.2 TWh for nickel), through there is much greater manufacturing capacity using global reserves (94 TWh for cobalt and 157 TWh for nickel)[29]. These values assume an NMC-811 battery chemistry. Materials constraints may be lessened through improvements in battery energy density, vehicle efficiency (decreased energy storage need), battery recycling[30], and the growth of alternative (e.g., LFP) and future (e.g., Na-ion) battery chemistries[31].

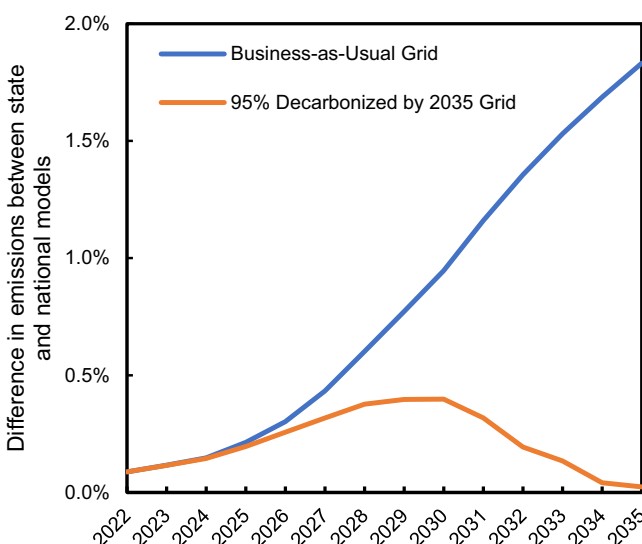

**Fig. 6 | Difference in greenhouse gas emissions between national and state models.** Percentage difference between the total light duty vehicle annual emissions when using state emissions factors or national average emission factors, for the business as usual (blue) and 95% decarbonization by 2035 (orange) grid scenarios.

## Additional policies

Under Section 177 of the Clean Air Act, California can set more stringent emissions standards than the Federal government. Currently 15 states (including California) have adopted California's Zero Emissions Vehicle (ZEV) goal of 67% EV sales by 2030 (exceeding the Federal goal of 50%)[16]. In our base scenario, the projected average among these 15 states is 64.4%. In this scenario California and Washington exceed (79% and 72%, respectively), Oregon meets (67%), and Nevada and Colorado are very close to this goal (65%, 66%), while the other CAA 177 states fall short. Here we conduct a sensitivity analysis, with the CAA states projected to exceed the goal staying the same as our base case, and the CAA states projected to fall short of the goal meeting it exactly (Supplemental Note 1).

This results in national EV sales increasing from 50% to 52% in 2030 (7.6 million vehicles to 8.0 million vehicles). In 2035 EV sales increase from 89% to 90% (13.4 to 13.5 million vehicles). The impact on EV stock percentage is less than 1% each year. The impact on GHG emissions is less than 1% in 2030 and approximately 1% in 2035, in both grid scenarios. These results suggest that a) Federal goals already rely on some states having more stringent policies than the Federal government, and b) current EV sales percentages (what we use to build our model) are a reasonable proxy for the differing levels of stringency in state level policies.

## Additional strategies

In addition to vehicle electrification and grid decarbonization, other strategies have been suggested to reduce LDV emissions including decreasing vehicle size[32], retiring vehicles faster than the natural fleet turnover rate[33, 34], and limiting or reducing LDV transport demand[35].

As we've shown, the current trend away from cars and towards trucks increases overall LDV fleet emissions by a couple percentage points. Halting this trend, or reversing it, would lead to a comparable emissions reduction. Additionally, both cars and trucks have been getting larger within their classes[17]. Incentivizing smaller, lighter vehicles could decrease emissions and may have co-benefits[36].

To investigate the potential of early retirement we test two potential early retirement policies (ER1 and ER2). These policies are implemented between 2025 and 2030 and involve gradually reducing the maximum vehicle age down to 20 years (ER1) or 15 years (ER2). For each policy we also show two pathways. In the first option, the annual

VMT schedule by vehicle age is maintained. Because new vehicles are driven more than old vehicles, this results in an increase in fleetwide VMT. In the second option, fleetwide VMT is held constant from the natural retirement scenario, which requires a decrease in annual per vehicle VMT. In each policy (ER1 and ER2) and each pathway (increased fleet VMT, constant fleet VMT) the overall number of vehicles is kept constant from the natural retirement scenario, which requires increased vehicle sales policy post-implementation (see Methods).

Each early retirement policy accelerates fleet turnover, resulting in an increase in the EV stock percentage reached by 2035 (Fig. 7a) and an increase in the fleetwide average fuel economy (Fig. 7b). However, due to increased sales (Fig. 7c) the production emissions also increase under these policies (Fig. 7d). The impact on total emissions depends upon how the retirement policy impacts VMT. If the addition of new vehicles results in an increase in fleetwide VMT, then total emissions may increase. However, if fleetwide VMT is reduced or remains the same, then the total emissions will decrease. As seen in Fig. 7e, early retirement policies initially increase emissions at the time of implementation, due to the increased production of replacement vehicles, but emissions are reduced over time due to lower operating phase emissions.

A final strategy for reducing LDV emissions is reducing VMT. This could be accomplished by reducing the number of vehicles or by reducing the miles traveled per vehicle. If reductions in the number of vehicles were spread evenly across the fleet, then emissions reductions would be proportional to VMT reductions (i.e., a 20% decrease in VMT results in a 20% decrease in emissions). If the miles traveled per vehicle is reduced, but the number of vehicles is not, then the emissions reductions are slightly less than proportional to the reduction in VMT (as production emissions are not reduced). For example, a 20% reduction in VMT results in a 17−18% reduction in emissions across the scenarios modeled here. Reductions may come from a decrease in demand (e.g., from teleworking) or through mode shifting to less intensive modes of transport (e.g., biking).

## Meeting targets

In Fig. 8 we combine hypothetical reductions using combinations of strategies discussed in this paper. For vehicle electrification we include 2030 sales of 50% (meeting current goals) and 67% (as if California's target were adopted nationally). For grid decarbonization we include the business-as-usual scenario and the 95% decarbonized by 2035 scenario as shown earlier. For early retirement we include our base case (natural retirement) and scenario ER2 with constant fleetwide VMT. And we combine this with our base case for VMT and a 20% reduction in VMT per vehicle. The total vehicle stock and fleetwide VMT, which change each year, are constant across all scenarios in Fig. 8. None of the scenarios explored achieve a 50% reduction from 2005 levels (800 Mt $CO_2e$) by 2030, though many combinations of strategies reach this goal by 2035.

## Discussion

Reducing the emissions of the U.S. LDV fleet and meeting decarbonization targets will require combining many different strategies[37]. To do so, the U.S. will need to:

- maintain aggressive vehicle electrification targets
- pair these targets with rapid grid decarbonization

However, while both are essential to long term transportation decarbonization, even in tandem they are insufficient to reach 800 Mt by 2030. Reaching short term goals on time will require additional strategies. Reductions of a few percent each are possible from:

- reducing vehicle production emissions
- reducing vehicle size (reducing size within classes and shifting to smaller classes)
- improving ICEV and EV fuel economies

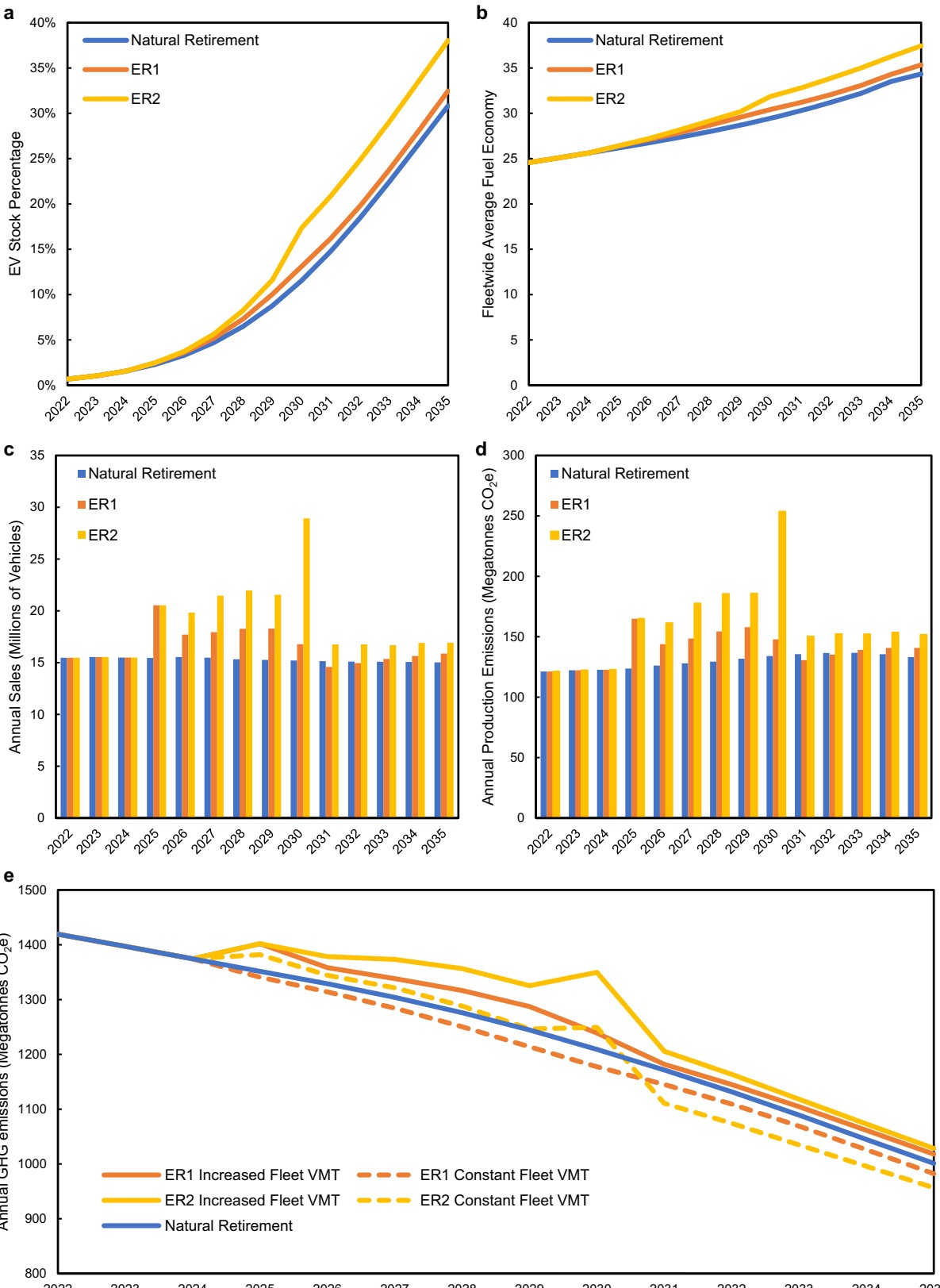

**Fig. 7 | Results of early vehicle retirement scenarios.** Annual impact on (**a**) EV stock percentage, (**b**) fleetwide average fuel economy, (**c**) vehicle sales, (**d**) vehicle production greenhouse gas emissions, and (**e**) greenhouse gas emissions (production and operation) for natural retirement (blue), early retirement 1 (ER1) (orange), and early retirement 2 (ER2) (yellow) scenarios. In **e**) the solid lines for ER1 and ER2 represent scenario in which per vehicle VMT is held constant, leading to an increase in fleet VMT from the natural retirement scenario, while the dashed lines represent scenarios in which fleet VMT is held constant with the natural retirement scenario.

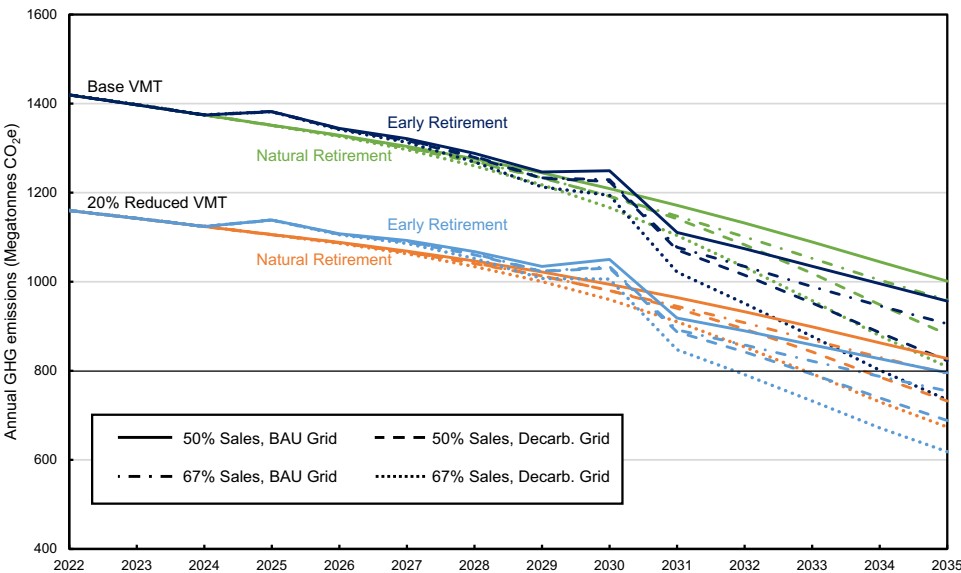

**Fig. 8 | Potential greenhouse gas emissions pathways for the light duty vehicle sector.** Greenhouse gas emissions pathways with different 2030 EV sales percentages (50%, 67%), different grid scenarios (business-as-usual (BAU) and 95% decarbonization by 2035 (Decarb. Grid)), different vehicle retirement policies (natural retirement and early retirement) and different levels of vehicle miles traveled (VMT) (base and 20% reduced). Dark blue represents early retirement with base VMT, green represents natural retirement with base VMT, light blue represents early retirement with 20% reduced VMT, and orange represents natural retirement with 20% reduced VMT.

As most short-term emissions reductions are attributable to fleet turnover, ICEV fuel economy standards still have an important role to play even during the transition to EVs. For the non-EV portion of the fleet, increased use of low carbon fuels[37] (not evaluated here) could reduce emissions for both new and existing ICEVs. Larger reductions may be achieved through policies that:

- reduce VMT (either through reductions in travel demand or shifting to less carbon intensive modes of transportation)
- accelerate fleet turnover through early retirement

Critically, all of the policies mentioned above should be integrated into a decarbonization strategy rather than considered individually[20]. For example, the impact of electrification is enhanced by grid decarbonization. The impact of early retirement is enhanced by more rapid electrification (as a greater percentage of the replacement vehicles are electric). Policies that retire vehicles early or reduce VMT would be more even impactful if they were targeted (e.g., reduce VMT specifically from ICEVs, or only retire vehicles early if they can be replaced with an EV).

Other trends and technologies not included in this analysis may contribute to changes in LDV emissions, including rebound effects[38], increased vehicle lifetimes[26], shared mobility services[39], the growth of ride-hailing fleets[40], connected and automated vehicles[41], micromobility technologies[42], vehicle-to-grid capabilities[43], charging behavior more generally[44], and heterogenous consumer preferences and behaviors[45, 46]. These have the potential to further reduce or increase the emissions of the LDV sector, and along with changes in vehicle size and power may change the socio-cultural expectations for vehicle performance[47]. Furthermore, there is significant uncertainty for many of the assumptions used for future technologies, including vehicle fuel economies, vehicle production emissions (especially for EV batteries), and grid decarbonization pathways.

The difference in our state-level modeling approach and our national fleet model reveals that national models slightly underestimate the emissions savings of electrification in the short term, but these approaches converge as the electric grid decarbonizes. Ultimately the difference in emissions between our state and national models is small compared to the scale of ICEV emissions in this time frame.

Transportation was responsible for 27% of U.S. greenhouse gas emissions in 2020[21] and was the single largest contributing sector. Light-duty vehicles are the largest sub-sector and are responsible for half of transportation emissions[3]. Here we show that the percentage of the reductions in LDV GHG emissions from 2022 levels that comes from fleet turnover is 80–87% in 2030 and 50–65% in 2035. Reaching a goal of 50% EV sales by 2030 would result in a decrease in emissions of 24% with business-as-usual grid carbon intensity, and 26% with a rapidly decarbonizing grid, compared to 2005 levels. This reflects the fact that in 2030 nearly 90% of LDV emissions would still be attributable to ICEVs. The significant delay between actions initiated in the present and GHG emissions reductions in the future reflects the time required for vehicle fleet turnover. Even if vehicle electrification and grid decarbonization goals are met or surpassed, the reduction in emissions from the LDV fleet (approximately 25% in 2030) would not meet a proportionate share of the national economy-wide emissions reduction goal of 50–52%. Light-duty vehicles are the largest contributor and one of the least difficult transportation modes to decarbonize. The fact that LDVs are likely to fall well short of the 50–52% target for 2030 implies that transportation as a whole will also fall well short of the target. However, we show that the benefits of increasing EV sales and grid decarbonization compound and increase over time such that by 2035 a 50% reduction in emissions from 2005 levels is plausible if vehicle electrification and grid decarbonization goals are met, particularly if other emissions reduction strategies are pursued concurrently. Furthermore, as vehicle electrification and grid decarbonization trajectories continue after 2035 (Fig. 8), reduction in LDV emissions will accelerate towards longer-term carbon neutrality goals.

## Methods
Our model consists of two main components – a fleet model that determines the number and types of vehicles in each state, and an emissions model that determines the life cycle greenhouse gas emissions of the vehicle fleet.

## Fleet model

The fleet model is based on the historical vehicle stock[27], projected vehicle sales[48], and vehicle scrappage rates[49] (Supplementary Fig. 1–3). The number of vehicles in the fleet (and of each model year) is determined by the number of vehicles in the fleet in the year prior, plus projected sales, minus the scrapped vehicles determined by the vehicle survival curves.

$$N_{y,p,c} = N_{y-1,p,c} + Sales_{y,p,c} - Scrapped_{y,p,c} \qquad (1)$$

where $y$ is the vehicle year, $p$ is the vehicle powertrain, and $c$ is the vehicle class. For powertrains we use ICEVs and EVs, where ICEV represents the sum of ICEV-gas, ICEV-diesel, CNG, FCV, HEV, and PHEVs, and EV represents the sum of 100-mile, 200-mile, and 300-mile range EVs. For classes we use cars and trucks, as defined in the 2022 Annual Energy Outlook sales projections[48].

## State fleets

To set the initial conditions for the state fleets, we need the percentage of car sales that were electric in 2021 and the percentage of truck sales that were electric in 2021 in each state. We approximate this data by first taking the percentage of LDV sales that were electric in each state, $\frac{EV\,Vehicles}{Vehicles}_s$, from the Alliance for Automotive Innovation[50], the number of cars and trucks relative to the total number of vehicles in each state, $\frac{Cars}{Vehicles}_s$ and $\frac{Trucks}{Vehicles}_s$, from vehicle registration data, and the number of EV cars and trucks compared to the total number of EVs nationally, $\frac{EV\,Cars}{EV\,Vehicles}$ and $\frac{EV\,Trucks}{EV\,Vehicles}$, from 2021 sales data, and solving for our desired initial values, $\frac{EV\,Cars}{Cars}_s$ and $\frac{EV\,Trucks}{Trucks}_s$, in each state using the system of equations shown below:

$$\frac{Cars}{Vehicles_s} * \frac{EV\,Cars}{Cars}_s + \frac{Trucks}{Vehicles_s} * \frac{EV\,Trucks}{Trucks}_s = \frac{EV\,Vehicles}{Vehicles}_s \qquad (2)$$

$$\frac{\frac{EV\,Cars}{Cars}_s}{\frac{EV\,Trucks}{Trucks}_s} = \frac{\frac{EV\,Cars}{EV\,Vehicles}}{\frac{EV\,Trucks}{EV\,Vehicles}} \qquad (3)$$

While imperfect, these data are combined to give estimates of the percentage of cars that are electric and the percentage of trucks that are electric in 2021, in each state.

We assume that the adoption rate is the same for cars and trucks and is the same in every state. The initial condition (percent electric in 2021) leads to different outcomes for cars and trucks and within each state as seen in Fig. 1 a, b. This also accounts for the fact that car-to-truck ratios vary greatly from over 2 in Washington DC to less than 0.5 in Alaska. We solve for the adoption rate, $r$, using a simple logistic function such that across all 50 states the total percentage of EV sales is exactly 50% in 2030:

$$P_{c,s}(2030) = \frac{K * P_{0,c,s}\,e^{r(2030)}}{K + P_{0,c}\left(e^{r(2030)} - 1\right)} \qquad (4)$$

where $P_{c,s}(2030)$ represents the percentage of cars that are electric in each state, s, in 2030. The carrying capacity, K, is 1 (corresponding to a maximum sales percentage of 100% electric vehicles). The initial condition, $P_{0,c,s}$, is the percentage of cars sales, c, that were electric in each state, s, in 2021.

$$P_{0,c,s} = \left(\frac{EV\,Cars}{Cars}\right)_s \qquad (5)$$

This is also done with trucks in every state:

$$P_{t,s}(2030) = \frac{K * P_{0,t,s}\,e^{r(2030)}}{K + P_{0,t,s}\left(e^{r(2030)} - 1\right)} \qquad (6)$$

where $P_{t,s}(2030)$ represents the percentage of trucks that are electric in each state, s, in 2030. The initial condition, $P_{0,t,s}$, is the percentage of truck sales, t, that were electric in each state, s, in 2021.

$$P_{0,t} = \left(\frac{EV\,Trucks}{Trucks}\right)_s \qquad (7)$$

We then take the summation of the percentage of cars that are electric in each state multiplied by that state's proportion of the total number of cars in the country, and the percentage of trucks that are electric in each state, multiplied by that state's proportion of the total number of trucks in the country, and set that equal to are sales target of 50% (or any other goal):

$$\sum_{s=1}^{51} P_{c,s}(2030) * \left(\frac{Cars_s}{Cars\,Nationally}\right) + P_{t,s}(2030) * \left(\frac{Trucks_s}{Trucks\,Nationally}\right) = 50\% \qquad (8)$$

This assumes that each state's proportion of the total number of vehicles in the country is constant throughout the study period. This system of equations can then be solved for the percentage of car and truck sales that are electric in each state in 2030, $P_{c,s}(2030)$ and $P_{t,s}(2030)$, and the required growth rate to reach those values, $r$. An EV stock comparison using linear rather than logistic growth for the adoption curves is shown in Supplementary Fig. 4.

## Emissions model

Once the number of vehicles (of each class, powertrain, and age) in each state has been determined, the use phase emissions can be calculated by multiplying the number of vehicles by the VMT (per vehicle), and by the greenhouse gas intensity of travel (per mile). These factors (number of vehicles, VMT, fuel economy, and grid intensity) can vary based on powertrain, vehicle class, state, and year.

$$LDV\,Fleet\,Use\,Phase\,Emissions = N_{p,c,s,y} * VMT_{p,c,s,y} * I_{p,c,s,y} \qquad (9)$$

where $N$ is the number of vehicles, $VMT$ is the vehicle miles traveled, $I$ is the GHG intensity of travel, $p$ is the powertrain, $c$ is the vehicle class, $s$ is the state and $y$ is the year. Not every component of the equation varies with all four indices. For example, we use VMT profiles that vary based on vehicle class and vehicle age but are the same across powertrains and states (Supplementary Fig. 5). The assumption that annual VMT is the same for ICEVs and EVs of the same class is justified by Gohlke and Zhou (2021), which shows that while short range EVs (< 150 miles) may be driven less than ICEVs, EVs with longer ranges have been found to drive as much or more as a comparable ICEV[51]. For a 300-mile range EV (near the U.S. average for new sales) the estimated EV and ICEV annual VMTs are within 5% of each other[51].

For ICEVs the GHG intensity of travel is determined by the life cycle carbon intensity of the fuel, $CI_F$, in kg $CO_2$e/gallon and the fuel economy, $FE_{ICEV}$, in miles/gallon.

$$I_{ICEV} = \frac{CI_F}{FE_{ICEV}} \qquad (10)$$

For EVs the GHG intensity of travel determined by the fuel economy, $FE_{EV}$, in Wh/mile and the grid emissions factor, $EF$, in kg $CO2_2$e/MWh, with appropriate unit conversions.

$$I_{BEV} = FE_{EV} * EF \qquad (11)$$

## Fuel economy

Fuel economy data comes from the VISION Model[52]. We take the weighted average (by market share) of 100-mile, 200-mile, and 300-

mile range EVs to determine the annual fuel economy of EV cars and EV trucks for each year. We divide the EV fuel economy, in Wh/mile, by 0.88 to account for charger efficiency. We use a similar weighted average for ICEVs, which consists of ICEV-gas, ICEV-diesel, CNG, FCV, HEV, and PHEV cars and trucks (Supplementary Fig. 6). Any increase in alternative vehicle sales (e.g., FCV, PHEV) is therefore reflected in the improved ICEV average fuel economy. For both ICEV and EV fuel economies we use the on-road correction factors from the Vision Model[52]. We also take a weighted average carbon intensity of the fuels for each year, corresponding to the different fleet mixes of each year, using data from GREET[53]. This includes combustion emissions and upstream emissions.

**Grid emissions**

We use grid emissions projections from NREL's Cambium model[28] for each state, as seen in Fig. 1. For Alaska and Hawaii we use projections from Vega-Perkins et al.[54]. These projections divide Alaska and Hawaii into two subregions each, so we use the average weighted by energy output from each subregion. We use two different emissions scenarios – a business as usual scenario (called the midcase within Cambium), and a 95% decarbonization by 2035 scenario (compared to 2005 levels). We use the combined combustion and upstream GHG emissions of electricity demand (not generation) in each state ($CO_2$, $CH_4$, and $N_2O$, 100-year GWP). Transmission and distribution losses are included in the Cambium values.

**Vehicle production emissions**

We obtain vehicle production emissions for model year 2020 and model year 2030 vehicles from Woody et al., (2022)[11]. We translate the three vehicle classes in that study (midsize sedan, midsize SUV, and pickup truck), into the two categories used in here (car and light truck), based on the 2021 sales ratios of the five categories used by the EPA[17] (Supplementary Fig. 7, Supplemental Note 2). We use linear interpolation to calculate vehicle production emissions specific to each year. Note that vehicle end-of-life emissions although relatively small are also included in the overall vehicle production emissions.

**Attribution analysis**

For the base case, vehicle electrification reaches 50% of sales in 2030, grid carbon intensity declines from 450 to 340 kg $CO_2$e/MWh between 2022 and 2030 ("business-as-usual"), ICEV average new vehicle fuel economy improves by approximately 3 MPG from 2022 to 2030, EV average new vehicle fuel economy is essentially unchanged from 2022 to 2030, the percentage of vehicles sold that are trucks increases from 67% to 72%, and the total number of vehicles sold annually declines by 1%. The VMT per vehicle annual schedule, which includes declining VMT as the vehicle ages, does not change throughout the study period. To construct Fig. 4, the emissions were calculated with a static truck sales percentage (67%), static EV sales percentage (approximately 4%), and static grid emissions (450 kg $CO_2$e/MWh). This shows the emissions reduction that will occur without these trends (i.e., due to fleet turnover). Fleet turnover captures the impact of improved fuel economy, the small change in the overall vehicle fleet size and most significantly, the replacement of older vehicles. The impact of truck percentage and electrification and grid improvement are calculated sequentially. Electrification and grid improvement are combined, as these trends interact strongly, as discussed in the Results. Reaching 50% EV sales increases the emissions due to vehicle production. The increase in vehicle production emissions shown in Fig. 4 represents the additional emissions from reaching the 50% EV sales target, relative to the static scenario in which EV sales remain at 4%.

**Early retirement**

We compare two different early retirement scenarios with our base case (natural retirement). In each scenario the early retirement policy is implemented between 2025 and 2030. In early retirement scenario 1 (ER1), the maximum vehicle age in 2025 is set at 25 years, and this decreases by one year for each year in the implementation period until the maximum vehicle age is 20 years in 2030. In early retirement scenario 2 (ER2) the age limit decreases by two years for each year in the implementation period, until the maximum vehicle age is 15 years in 2030. The maximum vehicle age is then kept constant after 2030. The overall number of vehicles is kept constant from the natural retirement scenario, so each early retirement scenario involves increased vehicle sales post-implementation.

For both ER1 and ER2 we investigate two potential pathways. In each pathway we maintain a consistent vehicle stock from the natural fleet turnover scenario (i.e., an additional vehicle is sold for each vehicle scrapped before its natural end of life). In pathway 1 we leave the annual VMT per vehicle schedule unchanged, which results in an increase in fleet VMT, as newer vehicles have a higher annual VMT than older vehicles. In pathway 2 we keep the annual fleetwide VMT constant. This is accomplished by decreasing the annual mileage per vehicle evenly amongst the entire vehicle fleet, so that the total fleetwide VMT remains the same. These two pathways represent two potential outcomes from a scrappage program, one in which fleetwide VMT increases due to the scrappage program and one in which fleetwide VMT is unchanged by the scrappage program. We do not investigate a scenario in which vehicle scrappage leads to a decrease in the number of vehicles or the fleetwide VMT (through this would be possible if scrapped vehicles are replaced with other modes of transit). If scrapped vehicles and their associated miles traveled were not replaced with new vehicles there could be an even greater decarbonizing effect.

**Reporting summary**

Further information on research design is available in the Nature Portfolio Reporting Summary linked to this article.

## Data availability

The data generated in this study are provided in the Supplementary Information/Source Data files. Source data are provided with this paper.

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

## Acknowledgements
The authors thank colleagues at Ford Motor Company (James Anderson, Robert De Kleine, Hyung Chul Kim, and Timothy Wallington) for helpful discussions. This work was supported by Ford Motor Company through a Ford-University of Michigan Alliance Project Award (M.W., G.A.K., P.V.) The findings and views expressed in this document are those of the authors and do not necessarily reflect the views of Ford Motor Company.

## Author contributions
M.W., conceptualization, methodology, investigation, formal analysis, visualization, writing-original draft, writing-review and editing; G.A.K., conceptualization, methodology, writing-review and editing, supervision; P.V., conceptualization, methodology, writing-review and editing.

## Competing interests
The authors declare no competing interests.
