## [Peer Review File · Nature Communications]

Decarbonization Potential of Electrifying 50% of U.S. Light-Duty Vehicle Sales by 2030Reviewers' Comments:

Reviewer #1:

Remarks to the Author:

his paper aims to assess the feasibility of the U.S. federal government's goals for vehicle electrification and economy-wide greenhouse gas (GHG) emissions reduction. Utilizing vehicle fleet and life cycle emissions models, we project future vehicle sales, stock, and emissions, taking into account state-level variations in electric vehicle adoption and grid emissions factors. By 2030, GHG emissions from the light-duty vehicle fleet can potentially be reduced by around 25% compared to 2005 levels, primarily through fleet turnover. However, to achieve emissions reductions approaching 45% by 2035, it is essential to meet both vehicle electrification and grid decarbonization goals. Accomplishing climate goals necessitates accelerated grid decarbonization alongside a transition to electric vehicles.

The exercise mentioned by the authors is highly valuable, as it adopts a bottom-up approach, focusing on vehicle sales, fleet composition, and GHG outcomes, rather than a top-down approach of determining the number of vehicles required for a specific GHG scenario. However, the demand model employed in this study has some notable flaws. It fails to consider the impact of policies such as Section 177 (i.e., states that joined California ACC 2) or the proposed changes in EPA CAFE regulations. Additionally, the model lacks a limitation on the total supply of vehicles per year, and the starting number of vehicles is the only determining factor, with the final numbers already pre-assigned to reach 100% by around 2040. This kind of model could be described as a combination of top-down and bottom-up approaches, and it would benefit from incorporating a maximum supply limitation per year, which would redistribute vehicles from non-177 states to 177 states. Furthermore, the model should differentiate the impact of plug-in hybrid electric vehicles (PHEVs) on cars and trucks and model them separately.

Moreover, the fleet model overlooks the possibility of early retirement for newer vehicles and the total end-of-life based on a certain mileage or number of years. As we move beyond 2030, the increasing market share of new EVs will replace retiring EVs, not just internal combustion engine (ICE) vehicles.

To enhance the analysis, I recommend incorporating scenarios into the EV adoption model that reflect the uneven distribution of new vehicles in states with stricter regulations and incentives. Additionally, the model should account for the varying impact of large and small vehicles, limiting the supply by restricting the number of batteries available each year.

Overall, the emission analysis alone does not present a novel contribution. Without a more robust and refined forecasting model, the paper's contribution remains limited and resembles other publications that predominantly focus on a top-down approach and nationwide analysis.

Reviewer #2:

Remarks to the Author:

This is a well executed, though not particularly innovative paper. Translating vehicle sales percentage projections into stock numbers is fairly well trodden ground. But the effort here is interesting, with the combination of stock modeling and grid decarbonization modeling at a state level. The various sensitivity cases are also useful. The combination of rapid EV uptake and rapid grid decarbonization case more or less eliminate the emissions problem from vehicles by 2040.

All the elements included in the analysis make sense to me and I really only have one significant comment on the paper: the state level assumptions regarding EV uptake are entirely based on their EV market shares in the base year. There is actually an important policy that could further differentiate these, which is the California Advanced Clean Cars II program, and its application to something like 15 other states under the CAA 177 program. This could push ZEV adoption to 67% by 2030 in these

states, which I think is around 40% of the US vehicle market. It would be very useful to see this scenario at least as a sensitivity case, if you assume all other states remain in their current positions as the base case.

Three other minor comments:

Probably when this paper was submitted, the EPA had not yet issued their proposed rule that will require a certain rate of CO₂ reduction to 2032, that they expect will have a particular impact on EV uptake, basically that it will hit Biden's 2030 target. They do also expect EV sales shares to hit 67% by 2032. These seem fairly consistent with your modeling (though not sure about the fuel economy part), but it would be useful to compare your scenarios to the EPA proposal, acknowledging that it is only a proposal at this point. Probably not worth changing your actual modeling given it is only a proposal, though if you thought something is likely to be significantly different as a result, you might consider doing another sensitivity case to account for it.

Please add a bit more information on VMT/vehicle. Figure 2b shows that newer vehicles are assumed to travel more per year, thus having a higher weight on overall emissions, but there is little discussion of this in relation to EVs vs ICE vehicles, for example can we expect new EVs to have the same VMT/vehicle as new ICE vehicles (given range limitations)? Or possibly higher than today, if people who buy new EVs choose to drive them intensively rather than other owned cars. Maybe worth a sensitivity case?

Finally, the spectrum of EV uptake rates by state based on where they are now, and the translation through to a 11% EV stock in 2030, even with sales at 50%, is surprising as I would have guessed it would be more like 20% or 25%. This seems to be a function of the S-curve shape and late, rapid sales increase toward 2030. I guess its also due to the long life of LDVs these days. Adding a comment on the nature of the adoption curve on stock changes would be useful, in that regard.

I found no typos or other minor issues, the paper seems very clean. The figures are all easily interpreted and well constructed.

Reviewer #3:

Remarks to the Author:

The paper uses sound methods and data sources to provide a top down accounting of transportation decarbonization, accounting for state level EV adoption and state level grid emissions. Providing state level resolution is useful and often not described in depth in other papers (examples from DOE/national labs below). However, the findings of the paper do not appear to be novel and the conclusions are not ground breaking. Transportation electrification does require also decarbonizing the electric sector.

<https://www.energy.gov/sites/default/files/2023-01/the-us-national-blueprint-for-transportation-decarbonization.pdf>

<https://www.nrel.gov/analysis/electrification-futures.html>

<https://www.nrel.gov/docs/fy22osti/81644.pdf>

The paper should better explain why this analysis is necessary or novel relative to prior analyses, who the intended audience is and what that audience should do with the results.

In addition, the paper should better reflect recent federal and state policies that will encourage electrification of the light duty transportation sector that could go beyond the 50% target of new light duty sales by 2030.

REVIEWER COMMENTS

Reviewer #1 (Remarks to the Author):

This paper aims to assess the feasibility of the U.S. federal government's goals for vehicle electrification and economy-wide greenhouse gas (GHG) emissions reduction. Utilizing vehicle fleet and life cycle emissions models, we project future vehicle sales, stock, and emissions, taking into account state-level variations in electric vehicle adoption and grid emissions factors. By 2030, GHG emissions from the light-duty vehicle fleet can potentially be reduced by around 25% compared to 2005 levels, primarily through fleet turnover. However, to achieve emissions reductions approaching 45% by 2035, it is essential to meet both vehicle electrification and grid decarbonization goals. Accomplishing climate goals necessitates accelerated grid decarbonization alongside a transition to electric vehicles.

The exercise mentioned by the authors is highly valuable, as it adopts a bottom-up approach, focusing on vehicle sales, fleet composition, and GHG outcomes, rather than a top-down approach of determining the number of vehicles required for a specific GHG scenario.

Thank you.

However, the demand model employed in this study has some notable flaws. It fails to consider the impact of policies such as Section 177 (i.e., states that joined California ACC 2) or the proposed changes in EPA CAFE regulations. Additionally, the model lacks a limitation on the total supply of vehicles per year, and the starting number of vehicles is the only determining factor, with the final numbers already pre-assigned to reach 100% by around 2040. This kind of model could be described as a combination of top-down and bottom-up approaches, and it would benefit from incorporating a maximum supply limitation per year, which would redistribute vehicles from non-177 states to 177 states.

There are three separate policies to discuss: the national sales goal of 50% EVs by 2030, the proposed EPA regulations that would result in 67% sales by 2032, and the states that have adopted the ZEV target of CAA 177, which is 67% sales by 2030. Ultimately, the base case we present using our model is consistent with all three of these policies.

First is the national sales goal of 50%. This is the goal that defines our base scenario (i.e., we investigate what the emissions would be if this goal were reached exactly).

The next policy is the proposed changes to EPA regulations, which would result in 67% EV sales by 2032. Because of the vehicle adoption curves that we use, our base scenario also aligns with (slightly exceeds) this goal. In our base scenario national EV sales reach 69% in 2032. We have added the following text to the Introduction to highlight that our modeling is consistent with this proposed policy (new text in blue):

"We use a top-down approach based on stated goals and proposed regulations (50% electrification of sales by 2030; 67% by 2032) along with current trends to determine

what the emissions would be if those goals are met. Our base scenario is defined by reaching exactly 50% EV sales nationwide in 2030. This results in 69% EV sales nationwide in 2032, so our base scenario is consistent with both stated goals (50% in 2030)⁶ and proposed EPA regulations (67% in 2032)⁷ .”

The third policy is CAA 177. This consists of 18 states (including California), however three have adopted the low emissions vehicle goals (which relate to fleet average fuel economy), without adopting the zero emissions vehicle goals (which include sales percentage targets for ZEVs). The 15 states (including California) that have adopted California’s ZEV targets have a goal of 67% sales by 2030 and 100% by 2035. Our model captures state-to-state variability, with some states reaching this target and others falling short. Of the 15 states, California and Washington exceed (79% and 72%, respectively), Oregon meets (67%), and Nevada and Colorado are very close to the 2030 goal (65%, 66%), while the other CAA 177 states fall short. In our base case (defined by 50% national sales in 2030) the weighted average across the CAA states is within 3% of the 67% goal (64.4%). It is also close to the 2035 goal (94.2% weighted average among CAA states). We have added the following text to the Introduction to clarify that our modeling is approximately consistent (on average) with the states that have adopted the CAA 177 ZEV policy.

“There are currently 15 states that have adopted a more aggressive sales goal of 67% by 2030 (rather than the national goal of 50%), and 100% by 2035 under the Clean Air Act Section 177. Our base model results in a weighted average EV sales percentage of 64.4% by 2030 and 94.2% by 2035 in these states, showing that our base case reasonably captures multiple Federal policies (Biden Administration goals and proposed EPA regulations) and state policies (ZEV goals under CAA 177).”

We have also added a sensitivity analysis in which the states that are projected to exceed the 67% ZEV sales goal (i.e., California and Washington) remain the same, and states projected to fall short of that goal instead reach it exactly (states that have not adopted California’s ZEV goal are also unchanged).

“Additional Policies

Under Section 177 of the Clean Air Act, California can set more stringent emissions standards than the Federal government. Currently 15 states (including California) have adopted California’s Zero Emissions Vehicle (ZEV) goal of 67% EV sales by 2030 (exceeding the Federal goal of 50%). In our base scenario, the projected average among these 15 states is 64.4%. In this scenario California and Washington exceed (79% and 72%, respectively), Oregon meets (67%), and Nevada and Colorado are very close to this goal (65%, 66%), while the other CAA 177 states fall short. Here we conduct a sensitivity analysis, with the CAA states projected to exceed the goal staying the same as our base case, and the CAA states projected to fall short of the goal meeting it exactly.

This results in national EV sales increasing from 50% to 52% in 2030 (7.6 million vehicles to 8.0 million vehicles). In 2035 EV sales increase from 89% to 90% (13.4 to 13.5 million vehicles). The impact on EV stock percentage is less than 1% each year. The impact on GHG emissions is less than 1% in 2030 and approximately 1% in 2035, in both grid

scenarios. These results suggest that a) Federal goals already rely on some states having more stringent policies than the Federal government, and b) current EV sales percentages (what we use to build our model) are a sufficient proxy for the differing levels of stringency in state level policies, in terms of vehicle sales, stock, and emissions outcomes.”

Because our base scenario is already a close approximation for targets based on CAA 177, and there is a small difference in results when CAA is met exactly, we believe an additional sensitivity analysis, in which CAA targets are met without increasing the national sales percentage (i.e., redistributing some EVs from non-CAA states to CAA states) would have a negligible impact on the results.

Furthermore, the model should differentiate the impact of plug-in hybrid electric vehicles (PHEVs) on cars and trucks and model them separately.

PHEVs are included in the ICEV fleet in our modeling. The higher fuel economy of PHEVs, and their increasing percentage as part of the fleet is reflected in the ICEV weighted average fuel economy, which increases from 38 to 44 MPG for cars and 27 to 29 MPG for trucks between 2022 and 2035. There is a small amount of error introduced by using MPGGE to approximate PHEV emissions, as the fuel cycle emissions would be slightly different. However, we are not convinced there is a significant benefit to modeling PHEVs separately, as most of their contribution is already included in the model through their fuel economy, and the PHEV sales are relatively small. We have added the following text to the methods section to clarify this point:

“We use a similar weighted average for ICEVs, which consists of ICEV-gas, ICEV-diesel, CNG, FCV, HEV, and PHEV cars and trucks (Fig. S5). Any increase in alternative vehicle sales (e.g., FCV, PHEV) is therefore reflected in the improved ICEV average fuel economy. There is a small amount of error introduced by using MPGGE to approximate PHEV emissions, as the fuel cycle emissions would be slightly different, but this effect will be small given the relatively low PHEV sales. For both ICEV and EV fuel economies we use the on-road correction factors from the Vision Model⁴⁸.”

Moreover, the fleet model overlooks the possibility of early retirement for newer vehicles and the total end-of-life based on a certain mileage or number of years. As we move beyond 2030, the increasing market share of new EVs will replace retiring EVs, not just internal combustion engine (ICE) vehicles.

We account for the fact that a small percentage of the vehicles retired each year will be EVs, and that this percentage will increase over time. As shown in our vehicle survival curves (Figure S3) approximately 2.5% of cars and 6% of trucks will be scrapped within the first five years of their lifetime. These survival curves are applied to EVs as well as ICEVs in our model.

As vehicle lifetimes have been increasing, rather than decreasing, any reduction in lifetime VMT would likely be the result of intentional policies to either incentivize early retirement or make scrappage mandatory after a certain number of years or miles.

To explore this possibility, we have added a section exploring two different early retirement policies and replaced Fig. 6 (now Fig. 7). In the results section:

“To investigate the potential of early retirement we test two potential early retirement policies (ER1 and ER2). These policies are implemented between 2025 and 2030 and involve gradually reducing the maximum vehicle age down to 20 years (ER1) or 15 years (ER2). For each policy we also show two pathways. In the first option, the annual VMT schedule by vehicle age is maintained. Because new vehicles are driven more than old vehicles, this results in an increase in fleetwide VMT. In the second option, fleetwide VMT is held constant from the natural retirement scenario, which requires a decrease in annual per vehicle VMT. In each policy (ER1 and ER2) and each pathway (increased fleet VMT, constant fleet VMT) the overall number of vehicles is kept constant from the natural retirement scenario, which requires increased vehicle sales policy post-implementation (see Methods).

Each early retirement policy accelerates fleet turnover, resulting in an increase in the EV stock percentage reached by 2035 (Fig. 7a) and an increase in the fleetwide average fuel economy (Fig. 7b). However, due to increased sales (Fig. 7c) the production emissions also increase under these policies (Fig. 7d). The impact on total emissions depends upon how the retirement policy impacts VMT. If the addition of new vehicles results in an increase in fleetwide VMT, then total emissions may increase. However, if fleetwide VMT is reduced or remains the same, then the total emissions will decrease. As seen in Fig. 7e, early retirement policies initially increase emissions at the time of implementation, due to the increased production of replacement vehicles, but emissions are reduced over time due to lower operating phase emissions.”

Figure 7. a) EV stock percentage, b) fleetwide average fuel economy, c) annual vehicle sales, d) annual vehicle production GHG emissions, and e) annual GHG emissions (production and operation) for natural retirement, early retirement 1 (ER1) and early retirement 2 (ER2) scenarios.

Additional details are provided in the Methods section:

“Early retirement

We compare two different early retirement scenarios with our base case (natural retirement). In each scenario the early retirement policy is implemented between 2025 and 2030. In early retirement scenario 1 (ER1), the maximum vehicle age in 2025 is set at 25 years, and this decreases by one year for each year in the implementation period until the maximum vehicle age is 20 years in 2030. In early retirement scenario 2 (ER2) the age limit decreases by two years for each year in the implementation period, until the maximum vehicle age is 15 years in 2030. The maximum vehicle age is then kept constant after 2030. The overall number of vehicles is kept constant from the natural retirement scenario, so each early retirement scenario involves increased vehicle sales post-implementation.

For both ER1 and ER2 we investigate two potential pathways. In each pathway we maintain a consistent vehicle stock from the natural fleet turnover scenario (i.e., an additional vehicle is sold for each vehicle scrapped before its natural end of life). In pathway 1 we leave the annual VMT per vehicle schedule unchanged, which results in an increase in fleet VMT, as newer vehicles have a higher annual VMT than older vehicles. In pathway 2 we keep the annual fleetwide VMT constant. This is accomplished by decreasing the annual mileage per vehicle evenly amongst the entire vehicle fleet, so that the total fleetwide VMT remains the same. These two pathways represent two potential outcomes from a scrappage program, one in which fleetwide VMT increases due to the scrappage program and one in which fleetwide VMT is unchanged by the scrappage program. We do not investigate a scenario in which vehicle scrappage leads to a decrease in the number of vehicles or the fleetwide VMT (through this would be possible if scrapped vehicles are replaced with other modes of transit). If scrapped vehicles and their associated miles traveled were not replaced with new vehicles there could be an even greater decarbonizing effect.”

We have also incorporated early retirement policies into our revised Figure 8:

“Meeting targets

In Fig. 8 we combine hypothetical reductions using combinations of strategies discussed in this paper. For vehicle electrification we include 2030 sales of 50% (meeting current goals) and 67% (as if California’s target were adopted nationally). For grid decarbonization we include the business-as-usual scenario and the 95% decarbonized by 2035 scenario as shown earlier. For early retirement we include our base case (natural retirement) and scenario ER2 with constant fleetwide VMT. And we combine this with our base case for VMT and a 20% reduction VMT per vehicle. The total vehicle stock and fleetwide VMT, which change each year, are constant across all scenarios in Fig. 8. None of the scenarios explored achieve a 50% reduction from 2005 levels (800 Mt CO₂e) by 2030, though many combinations of strategies reach this goal by 2035.”

Figure 8. Emissions pathways with different 2030 EV sales percentages (50%, 67%), different grid scenarios (business-as-usual (BAU) and 95% decarbonization by 2035 (Decarb. Grid)), different vehicle retirement policies (natural retirement and early retirement) and different levels of vehicle miles traveled (VMT) (base and 20% reduced).

To enhance the analysis, I recommend incorporating scenarios into the EV adoption model that reflect the uneven distribution of new vehicles in states with stricter regulations and incentives.

Though it is based on current sales percentages, rather than regulations and incentives, our base case does reflect the uneven distribution of new vehicles in different states. For example, in our base scenario, the EV sales percentage in 2030 is 16% in North Dakota, 37% in Michigan, 50% in New York, and 79% in California. We have added a new figure to clarify this point.

Figure 1. Projected EV sales percentage in each state in 2030, with a national average sales percentage of 50%

Additionally, as shown in our analysis on CAA 177, current sales percentages come fairly close to approximating the impact of variable state level policies. From our new section *Additional policies*:

“This results in national EV sales increasing from 50% to 52% in 2030 (7.6 million vehicles to 8 million vehicles). In 2035 EV sales increase from 89% to 90% (13.4 to 13.5 million vehicles). The impact on EV stock percentage is less than 1% each year. The impact on GHG emissions is less than 1% in 2030 and approximately 1% in 2035, in both grid scenarios. These results suggest that a) Federal goals already rely on some states having more stringent policies than the Federal government, and b) current EV sales percentages (what we use to build our model) are a reasonable proxy for the differing levels of stringency in state level policies.”

Additionally, the model should account for the varying impact of large and small vehicles, limiting the supply by restricting the number of batteries available each year.

An analysis of limitations based on battery supply is a fundamentally different type of question than what is pursued in this paper. Including a battery supply limitation would involve predicting the future availability of batteries. Our work here is not to predict the most likely future outcome, but rather to evaluate future outcomes based upon achieving stated policy goals (electrification and grid decarbonization targets).

The varying impact of small and large vehicles is shown in Fig. 5 (slight increase in emissions attributable to increasing truck percentage) and mentioned again in our section *Additional strategies*:

“As we’ve shown, the current trend away from cars and towards trucks increases overall LDV fleet emissions by a couple percentage points. Halting this trend, or reversing it, would lead to a comparable emissions reduction. Additionally, both cars and trucks have been getting larger within their classes¹⁶. Incentivizing smaller, lighter vehicles could decrease emissions and may have co-benefits³⁶.”

As far as how batteries are distributed between cars and trucks, changing this percentage (while maintaining a fixed supply of batteries in kWh) would not have a major impact on the results. While cars use smaller batteries, allowing more cars to be electrified from the same amount of battery materials, the savings per vehicle is greater for larger vehicles. Additionally, SUVs and trucks have greater lifetime mileages on average. Therefore, the lifetime savings per kWh of battery material is surprisingly similar - see figure below using values from Woody et al., 2022. If the distribution of batteries between cars and trucks leads to an effect on the sales percentage between cars and trucks, there may be an impact. However, determining whether or not powertrain or vehicle class is the larger determinant in vehicle purchasing decisions is well outside the scope of this paper.

Nevertheless, we agree that battery material constraints are an important factor to consider. We have added the following paragraph to our results section detailing the battery usage in our modeled scenario and comparing this with domestic and worldwide material reserves:

“Battery material constraints

Under our base scenario, the US EV stock reaches 11.5% in 2030 and 30.8% in 2035. This is approximately 31 million EV cars and 47 million EV trucks on the road in 2035. This would require roughly 7.5 TWh of batteries, using current vehicle battery sizes. This is roughly equal to the total battery manufacturing capacity using U.S. lithium reserves and well below the total manufacturing capacity with global lithium reserves (209 TWh)²⁸. The manufacturing capacity using U.S. reserves of other critical materials are more limited (0.7 TWh for cobalt and 0.2 TWh for nickel), through there is much greater manufacturing capacity using global reserves (94 TWh for cobalt and 157 TWh for nickel)²⁸. These values assume an NMC-811 battery chemistry. Materials constraints may be lessened through improvements in battery energy density, vehicle efficiency (decreased energy storage need), battery recycling²⁹, and the growth of alternative (e.g., LFP) and future (e.g., Na-ion) battery chemistries³⁰.”

Overall, the emission analysis alone does not present a novel contribution. Without a more robust and refined forecasting model, the paper's contribution remains limited and resembles other publications that predominantly focus on a top-down approach and nationwide analysis.

We believe there are many novel contributions of this work. To the best of our knowledge this is the first paper to:

- Quantify the gap between stated vehicle electrification goals and decarbonization goals
- Focus on shorter term goals (2030-2035) rather than 2050 targets
- Include state by state analysis rather than national analysis, incorporating both state level and national level EV targets
- Include attribution of emissions reduction to specific changes in the LDV sector
- Incorporate updated electric grid decarbonization scenarios that reflect more aggressive decarbonization timelines

Thank you for your comments. Your feedback has helped us make significant improvements to the paper, including additional analysis of CAA 177 and vehicle retirement policies, as well as improving the clarity of our work.

Reviewer #2 (Remarks to the Author):

This is a well-executed, though not particularly innovative paper. Translating vehicle sales percentage projections into stock numbers is fairly well trodden ground. But the effort here is interesting, with the combination of stock modeling and grid decarbonization modeling at a state level. The various sensitivity cases are also useful. The combination of rapid EV uptake and rapid grid decarbonization case more or less eliminate the emissions problem from vehicles by 2040.

Thank you.

All the elements included in the analysis make sense to me and I really only have one significant comment on the paper: the state level assumptions regarding EV uptake are entirely based on their EV market shares in the base year. There is actually an important policy that could further differentiate these, which is the California Advanced Clean Cars II program, and its application to something like 15 other states under the CAA 177 program. This could push ZEV adoption to 67% by 2030 in these states, which I think is around 40% of the US vehicle market. It would be very useful to see this scenario at least as a sensitivity case, if you assume all other states remain in their current positions as the base case.

This is an important point. Even though our state level values rely on EV market shares in the base year, this actually closely approximates CAA 177, with the weighted average among states that have adopted the ZEV goal having a 64.4% EV sales percentage in 2030 (compared to a goal of 67%). We have added the following language to the introduction to clarify this point (new text in blue):

“Modeling each state individually, rather than the country as a whole, is a novel contribution of our study that allows us to investigate the impact of state-level heterogeneity in EV adoption levels and grid emissions factors on overall LDV emissions. As the proportion of EVs in the fleet grows, where and when these vehicles charge will become more important¹⁵. There are currently 15 states that have adopted more a more aggressive sales goal of 67% by 2030 (rather than the national goal of 50%), and 100% by 2035 under the Clean Air Act Section 177. Our base model results in a weighted average EV sales percentage of 64.4% by 2030 and 94.2% by 2035 in these states, showing that the base case we present using our model reasonably captures multiple Federal policies (Biden Administration goals and proposed EPA regulations) and state policies (ZEV goals under CAA 177) simultaneously.”

We have added a sensitivity analysis in which the states that are projected to exceed the 67% ZEV sales goal remain the same, and states projected to fall short of that goal instead reach it exactly (states that have not adopted California’s ZEV goal are also unchanged):

“Additional Policies

Under Section 177 of the Clean Air Act, California can set more stringent emissions standards than the Federal government. Currently 15 states (including California) have adopted California’s Zero Emissions Vehicle (ZEV) goal of 67% EV sales by 2030 (exceeding the Federal goal of 50%)²⁸. In our base scenario, the projected average among these 15 states is 64.4%. In this scenario California and Washington exceed (79% and 72%, respectively), Oregon meets (67%), and Nevada and Colorado are very close to this goal (65%, 66%), while the other CAA 177 states fall short. Here we conduct a sensitivity analysis, with the CAA states projected to exceed the goal staying the same as our base case, and the CAA states projected to fall short of the goal meeting it exactly (Supplemental Note 1).

This results in national EV sales increasing from 50% to 52% in 2030 (7.6 million vehicles to 8 million vehicles). In 2035 EV sales increase from 89% to 90% (13.4 to 13.5 million vehicles). The impact on EV stock percentage is less than 1% each year. The impact on GHG emissions is less than 1% in 2030 and approximately 1% in 2035, in both grid scenarios. These results suggest that a) Federal goals already rely on some states having more stringent policies than the Federal government, and b) current EV sales percentages (what we use to build our model) are a sufficient proxy for the differing levels of stringency in state level policies, in terms of vehicle sales, stock, and emissions outcomes.”

Three other minor comments:

Probably when this paper was submitted, the EPA had not yet issued their proposed rule that will require a certain rate of CO2 reduction to 2032, that they expect will have a particular impact on EV uptake, basically that it will hit Biden's 2030 target. They do also expect EV sales shares to hit 67% by 2032. These seem fairly consistent with your modeling (though not sure about the fuel economy part), but it would be useful to compare your scenarios to the EPA proposal, acknowledging that it is only a proposal at this point. Probably not worth changing your actual modeling given it is only a proposal, though if you thought something is likely to be significantly different as a result, you might consider doing another sensitivity case to account for it.

In our modeling, the Biden 2030 target is consistent with (slightly exceeding) the expectation of the EPA proposed rule for 67% EV sales share by 2032. Due to the logistic adoption curves we used, our base scenario (50% sales in 2030) results in 69% EV sales in 2032.

The EPA proposed rule is first mentioned in the Introduction:

"Through an Executive order, the U.S. has set a non-binding target for 50% of LDV sales to be electric by 2030. New fuel economy standards proposed by the Environmental Protection Agency (EPA) may result in 67% of new LDV sales being electric by 2032. And some U.S. states, led by California, have a more ambitious target for 100% of LDV sales to be electric by 2035."

Later in the Introduction we add the following text to clarify the alignment between the Biden 2030 target and the proposed EPA rules:

"We use a top-down approach based on stated goals and proposed regulations (50% electrification of sales by 2030; 67% by 2032) along with current trends to determine what the emissions would be if those goals are met. Our base scenario is defined by reaching exactly 50% EV sales nationwide in 2030. This results in 69% EV sales nationwide in 2032, so our base scenario is consistent with both stated goals (50% in 2030)⁶ and proposed EPA regulations (67% in 2032)⁷."

In the Results section:

"We tune the growth rate such that the target value of 50% EV sales nationwide in 2030 is reached exactly. This results in 69% EV sales nationwide in 2032, so our model is consistent with both stated goals (50% in 2030)⁶ and proposed EPA regulations (67% in 2032)⁷."

Please add a bit more information on VMT/vehicle. Figure 2b shows that newer vehicles are assumed to travel more per year, thus having a higher weight on overall emissions, but there is little discussion of this in relation to EVs vs ICE vehicles, for example can we expect new EVs to have the same VMT/vehicle as new ICE vehicles (given range limitations)? Or possibly higher than today, if people who buy new EVs choose to drive them intensively rather than other owned cars. Maybe worth a sensitivity case?

In our base case we assume EVs have identical VMT/vehicle as ICEVs, as there is not a consensus on whether the VMT of EVs is more or less than ICEVs on average. We added the following citation to support this assumption (in the emissions modeling section of the Methods):

“For example, we use VMT profiles that vary based on vehicle class and vehicle age but are the same across powertrains and locations (Fig. S4). The assumption that annual VMT is the same for ICEVs and EVs of the same class is justified by Gohlke and Zhou (2021), which shows that while short range EVs (<150 miles) may be driven less than ICEVs, EVs with longer ranges have been found to drive as much or more as a comparable ICEV⁴⁸. For a 300-mile range EV (near the U.S. average for new sales) the estimated EV and ICEV annual VMTs are within 5% of each other⁴⁸.”

Finally, the spectrum of EV uptake rates by state based on where they are now, and the translation through to a 11% EV stock in 2030, even with sales at 50%, is surprising as I would have guessed it would be more like 20% or 25%. This seems to be a function of the S-curve shape and late, rapid sales increase toward 2030. I guess its also due to the long life of LDVs these days. Adding a comment on the nature of the adoption curve on stock changes would be useful, in that regard.

The long life of LDVs is the primary reason that the LDV stock only reaches 11% in 2030, even with 50% sales in that year. The S-curve shape also contributes but is not a huge factor. To compare with the logistic growth scenario, we have added a scenario in which EV sales grow linearly up to 50% in 2030, as a bounding exercise. In this case the LDV stock reached 15% in 2030. We have added the following text to the section *Vehicle sales, stock and emissions*:

“The 50% electrification rate for new vehicle sales in 2030 has a limited impact on the vehicle stock in that year, due to the time required for the LDV fleet to turn over. The median vehicle in the U.S. is on the road for approximately 20 years²⁵. Using vehicle survival curves from the Transportation Energy Data Book²⁶, we show that achieving a 50% electric sales rate in 2030 would lead to 11.5% of the LDV stock being electric in 2030. This percentage could be even lower if vehicle lifetimes continue to increase²⁵. The shape of the adoption curve has a limited impact; even if sales were to grow linearly to 50%, rather than logistically, only 15.1% of the LDV stock would be electric in 2030.

We have added the following sentence to the *State fleets* subsection of the Methods and the following figure to the supplemental information:

“An EV stock comparison using linear rather than logistic growth for the adoption curves is shown in Fig. S4.”

Figure S4. a) With logistic growth, 50% EV sales in 2030 results in 11% EV stock in 2030, b) With linear growth, 50% EV sales in 2030 results in 15% EV stock in 2030.

I found no typos or other minor issues, the paper seems very clean. The figures are all easily interpreted and well-constructed.

Thank you.

Reviewer #3 (Remarks to the Author):

The paper uses sound methods and data sources to provide a top down accounting of transportation decarbonization, accounting for state level EV adoption and state level grid emissions. Providing state level resolution is useful and often not described in depth in other papers (examples from DOE/national labs below). However, the findings of the paper do not appear to be novel and the conclusions are not ground breaking. Transportation electrification does require also decarbonizing the electric sector.

<https://www.energy.gov/sites/default/files/2023-01/the-us-national-blueprint-for-transportation-decarbonization.pdf>

<https://www.nrel.gov/analysis/electrification-futures.html>

<https://www.nrel.gov/docs/fy22osti/81644.pdf>

Thank you for your comments. Our finding isn't just that transportation decarbonization also requires decarbonizing the electric sector. Our results show that in the near term (until 2030), grid decarbonization will have a very minor impact on LDV emissions, because of the low percentage of electric vehicles in the fleet. Much of the decarbonization achieved in this time period is from the replacement of ICEVs in the fleet with new ICEVs with improved fuel economy. We also show that while grid decarbonization is more impactful on 2035 emissions, rapid vehicle electrification and electricity decarbonization are insufficient to meet transportation decarbonization goals, absent other actions. This includes some of the other strategies we mention in the paper, like decreasing VMT, decreasing the carbon intensity of vehicle production, shifts in vehicle size, and potentially early retirement of older vehicles.

The paper should better explain why this analysis is necessary or novel relative to prior analyses, who the intended audience is and what that audience should do with the results.

We believe there are many novel contributions of this work. To the best of our knowledge this is the first paper to:

- Quantify the gap between stated vehicle electrification goals and decarbonization goals
- Focus on shorter term goals (2030-2035) rather than 2050 targets
- Include state by state analysis rather than national analysis, incorporating both state level and national level EV targets
- Include attribution of emissions reduction to specific changes in the LDV sector
- Incorporate updated electric grid decarbonization scenarios that reflect more aggressive decarbonization timelines

We have added the following text to the introduction to clarify the novelty of this work:

“Modeling each state individually, rather than the country as a whole, is a novel contribution of our study that allows us to investigate the impact of state-level heterogeneity in EV adoption levels and grid emissions factors on overall LDV emissions.”

“...our base case reasonably captures multiple Federal policies (Biden Administration goals and proposed EPA regulations) and state policies (ZEV goals under CAA 177).”

We have significantly revised the first half of the Discussion section of the paper to expand upon the significance of our results for researchers and policy makers.

“Discussion

Reducing the emissions of the U.S. LDV fleet and meeting decarbonization targets will require combining many different strategies. To do so, the U.S. will need to:

- *maintain aggressive vehicle electrification targets*
- *pair these targets with rapid grid decarbonization*

However, while both are essential to long term transportation decarbonization, even in tandem they are insufficient to reach 800 Mt by 2030. Reaching short term goals on time will require additional strategies. Reductions of a few percent each are possible from:

- *reducing vehicle production emissions*
- *reducing vehicle size (reducing size within classes and shifting to smaller classes)*
- *improving ICEV and EV fuel economies*

As most short-term emissions reductions are attributable to fleet turnover, ICEV fuel economy standards still have an important role to play even during the transition to EVs. For the non-EV portion of the fleet, increased use of low carbon fuels³⁷ (not evaluated here)

could reduce emissions for both new and existing ICEVs. Larger reductions may be achieved through policies that:

- *reduce VMT (either through reductions in travel demand or shifting to less carbon intensive modes of transportation)*
- *accelerate fleet turnover through early retirement*

Critically, all of the policies mentioned above should be integrated into a decarbonization strategy rather than considered individually¹⁹. For example, the impact of electrification is enhanced by grid decarbonization. The impact of early retirement is enhanced by more rapid electrification (as a greater percentage of the replacement vehicles are electric). Policies that retire vehicles early or reduce VMT would be more even impactful if they were targeted (e.g., reduce VMT specifically from ICEVs, or only retire vehicles early if they can be replaced with an EV).”

We have also revised the following sentence near the end of the Discussion section:

“However, we show that the benefits of increasing EV sales and grid decarbonization compound and increase over time such that by 2035 a 50% reduction in emissions from 2005 levels is plausible if vehicle electrification and grid decarbonization goals are met, particularly of other emissions reduction strategies are pursued concurrently.”

In addition, the paper should better reflect recent federal and state policies that will encourage electrification of the light duty transportation sector that could go beyond the 50% target of new light duty sales by 2030.

We have added text and sensitivity analysis to reflect how our work relates to two different policies: the proposed EPA regulations and state level ZEV targets under CAA177. While our base case is defined by the national sales goal of 50% Evs by 2030, our base case is consistent these two additional policies.

The proposed changes to EPA regulations would result in 67% EV sales by 2032. Because of the vehicle adoption curves that we use, our base scenario aligns with (slightly exceeds) this goal. In our base scenario national EV sales reach 69% in 2032. We have added the following text to the introduction to highlight that our modeling is consistent with this proposed policy:

“We use a top-down approach based on stated goals and proposed regulations (50% electrification of sales by 2030; 67% by 2032) along with current trends to determine what the emissions would be if those goals are met. Our base scenario is defined by reaching exactly 50% EV sales nationwide in 2030. This results in 69% EV sales nationwide in 2032, so our base scenario is consistent with both stated goals (50% in 2030)⁶ and proposed EPA regulations (67% in 2032)⁷ .”

There are 18 states (including California) that have adopted some of California’s more stringent vehicle policies under section 177 of the Clean Air Act. Three have adopted the low emissions vehicle goals (which relate to fleet average fuel economy), without adopting the zero emissions vehicle goals (which include sales percentage targets for ZEVs). The 15 states (including California) that have adopted California’s ZEV targets have a goal of 67% sales by 2030. Our model captures state-to-state variability, with some states reaching this target and others falling short. Of the 15 states, California and Washington exceed (79% and 72%, respectively), Oregon meets (67%), and Nevada and Colorado are very close to this goal (65%, 66%), while the other CAA 177 states fall short. In our base case (50% nationally in 2030) the weighted average across the CAA states is within 3% of the 67% goal (64.4%). We have added the following text to the introduction to clarify that our modeling is approximately consistent (on average) with the states that have adopted the CAA 177 ZEV policy.

“There are currently 15 states that have adopted a more aggressive sales goal of 67% by 2030 (rather than the national goal of 50%), and 100% by 2035 under the Clean Air Act Section 177. Our base model results in a weighted average EV sales percentage of 64.4% by 2030 and 94.2% by 2035 in these states, showing that the base case we present using our model reasonably captures multiple Federal policies (Biden Administration goals and proposed EPA regulations) and state policies (ZEV goals under CAA 177) simultaneously.”

We have added an additional sensitivity analysis in which the states that are projected to exceed the 67% ZEV sales goal remain the same, and states projected to fall short of

that goal instead reach it exactly (states that have not adopted California's ZEV goal also are unchanged).

“Additional Policies

Under Section 177 of the Clean Air Act, California can set more stringent emissions standards than the Federal government. Currently 15 states (including California) have adopted California's Zero Emissions Vehicle (ZEV) goal of 67% EV sales by 2030 (exceeding the Federal goal of 50%). In our base scenario, the projected average among these 15 states is 64.4%. In this scenario California and Washington exceed (79% and 72%, respectively), Oregon meets (67%), and Nevada and Colorado are very close to this goal (65%, 66%), while the other CAA 177 states fall short. Here we conduct a sensitivity analysis, with the CAA states projected to exceed the goal staying the same as our base case, and the CAA states projected to fall short of the goal meeting it exactly (Supplemental Note 1).

This results in national EV sales increasing from 50% to 52% in 2030 (7.6 million vehicles to 8.0 million vehicles). In 2035 EV sales increase from 89% to 90% (13.4 to 13.5 million vehicles). The impact on EV stock percentage is less than 1% each year. The impact on GHG emissions is less than 1% in 2030 and approximately 1% in 2035, in both grid scenarios. These results suggest that a) Federal goals already rely on some states having more stringent policies than the Federal government, and b) current EV sales percentages (what we use to build our model) are a sufficient proxy for the differing levels of stringency in state level policies, in terms of vehicle sales, stock, and emissions outcomes.”

Finally, we have revised Fig. 8 and the supporting text to include a scenario in which EVs are 67% of sales by 2030 (as if California's target were adopted nationally):

“Meeting targets

In Fig. 8 we combine hypothetical reductions using combinations of strategies discussed in this paper. For vehicle electrification we include 2030 sales of 50% (meeting current goals) and 67% (as if California's target were adopted nationally). For grid decarbonization we include the business-as-usual scenario and the 95% decarbonized by 2035 scenario as shown earlier. For early retirement we include our base case (natural retirement) and scenario ER2 with constant fleetwide VMT. And we combine this with our base case for VMT and a 20% reduction VMT per vehicle. The total vehicle stock and fleetwide VMT, which change each year, are constant all scenarios in Fig. 8. None of the scenarios explored achieve a 50% reduction from 2005 levels (800 Mt CO₂e) by 2030, though many combinations of strategies reach this goal by 2035.”

Figure 8. Emissions pathways with different 2030 EV sales percentages (50%, 67%), different grid scenarios (business-as-usual (BAU) and 95% decarbonization by 2035 (Decarb. Grid)), different vehicle retirement policies (natural retirement and early retirement) and different levels of vehicle miles traveled (VMT) (base and 20% reduced).

Reviewers' Comments:

Reviewer #2:

Remarks to the Author:

I find the authors have done a good job addressing my one major and three minor points from the previous round of review. Other changes (additional scenarios, additional clarifying text) are also welcome. I have no further comments on the paper and support it for publication.

Reviewer #3:

Remarks to the Author:

The revisions are helpful and I appreciate the authors taking the time to respond to comments. I stand by my original comments that this analysis is consistent with current projections. However, the additional data at the state level is helpful. I have no concerns about publishing this paper.

Reviewer #2:

I find the authors have done a good job addressing my one major and three minor points from the previous round of review. Other changes (additional scenarios, additional clarifying text) are also welcome. I have no further comments on the paper and support it for publication.

Thank you for your review, which helped us to improve the paper.

Reviewer #3:

The revisions are helpful and I appreciate the authors taking the time to respond to comments. I stand by my original comments that this analysis is consistent with current projections. However, the additional data at the state level is helpful. I have no concerns about publishing this paper.

Thank you for your review, which helped us to improve the paper.